# Neonatal Jaundice Treatment Versus Recommendations: The Challenge of Treatment Without Rapid Diagnostic Capability

**DOI:** 10.3390/ijerph22071032

**Published:** 2025-06-28

**Authors:** Ashura Bakari, Ann V. Wolski, Benjamin Otoo, Rexford Amoah, Elizabeth Kaselitz, Sarah D. Compton, Rebekah Shaw, Cheryl A. Moyer

**Affiliations:** 1Department of Child Health, Suntreso Government Hospital, Ghana Health Service, Kumasi AK-039, Ghana; abakari@yahoo.com (A.B.); beotoo55@gmail.com (B.O.); rexfordrex@gmail.com (R.A.); 2Departments of Obstetrics and Gynecology, Pediatrics, and Learning Health Sciences, University of Michigan Medical School, Ann Arbor, MI 48109, USA; avwolski@gmail.com (A.V.W.); lbmaccorm@gmail.com (E.K.); sarahrom@umich.edu (S.D.C.); rebekash@med.umich.edu (R.S.)

**Keywords:** jaundice, newborn health, Africa

## Abstract

Neonatal jaundice (NNJ) is a leading cause of death in the early neonatal period, disproportionately affecting newborns in sub-Saharan Africa. In a setting without access to rapid assessment via transcutaneous bilirubin meter, we sought to determine how closely the diagnosis and treatment of neonatal jaundice at an urban district hospital aligned with retrospective assessment and treatment recommendations using the BiliApp (based on the UK NICE Guideline CG98). This retrospective chart review study aimed to identify: (1) What percent of admissions within 8 days of birth to the Mother and Baby Unit (MBU) at our study site were admitted for a primary diagnosis of neonatal jaundice, and what characterized those admissions? (2) How did treatment provided compare to the recommendations of the United Kingdom NICE Guideline CG98 via the “BiliApp”? and (3) Among those with jaundice, what factors were associated with an increased likelihood of severity indicative of the need for blood exchange therapy? The charts of all neonates admitted to the MBU at Suntreso Government Hospital (SGH), in Kumasi, Ghana, in 2020 were reviewed by trained research assistants. Data were collected regarding demographics, reason for admission, diagnostic markers (e.g., serum bilirubin level), treatments performed in the hospital, and outcome. Data were analyzed using Stata 18.0. There were 1059 newborns admitted to the MBU in 2020 at less than 8 days of age. A total of 179 (16.9%) were admitted with a primary diagnosis of neonatal jaundice. According to the BiliApp, 29.4% (*n* = 50) of newborns admitted for jaundice had bilirubin levels that were normal or below the phototherapy threshold for their gestational age on admission; 25.3% (*n* = 43) were at or near the threshold for phototherapy; 21.2% (*n* = 36) were above the phototherapy threshold; and 24.1% (*n* = 41) were above the blood exchange therapy threshold. The BiliApp recommended no treatment for 21.2% (*n* = 36) of newborns, repeated assessment for 33.6% (*n* = 57), phototherapy for 21.2% (*n* = 36), and exchange therapy for 24.1% (*n* = 41). By comparison, 8.2% (*n* = 14) of neonates admitted for jaundice received no treatment, 77.8% (*n* = 133) received phototherapy only, and 14.0% (*n* = 24) received both phototherapy and exchange therapy. Without sufficient data on G6PD status and parent/newborn blood type to include in the analysis, the biggest risk factors for a BiliApp recommendation of exchange therapy included serum bilirubin level (OR 1.01, *p* < 0.001) and gestational age (OR 0.51, *p* < 0.001), even after controlling for breastfeeding and male sex of the newborn. Without access to rapid assessment tools, many providers in low-resource settings are put in a position to presumptively treat newborns suspected of having jaundice, rather than waiting for serum lab tests to return. Given the cost of transcutaneous bilirubin meters, additional options for rapid diagnostic testing are warranted.

## 1. Introduction

Globally, neonatal jaundice (NNJ) is a leading cause of death in the neonatal period (0–28 days), with sub-Saharan Africa bearing a disproportionate burden of jaundice-related morbidity and mortality [1]. A recent meta-analysis showed that 7.5% of hospitalized neonates with jaundice in the African region died, compared to less than 1% in both Europe and the Americas [2]. In addition to the mortality burden, as unconjugated bilirubin can penetrate the blood–brain barrier in newborns, neonatal jaundice leads to long-term sequalae for up to 70% of survivors globally, including choreo-athetoid cerebral palsy, deafness, speech and language processing disorders, and learning difficulties [2]. NNJ, characterized by a yellowish discoloration of the skin and eyes, is a common, treatable condition in newborns [3]. It occurs when bilirubin—a byproduct of heme oxidation—accumulates in body tissues. Clinically apparent when the total serum bilirubin (TSB) level exceeds 5 mg/dL (85 micromol/L) [4], NNJ stems from an increased production of bilirubin, secondary to increased circulating red blood cells and a shortened red cell lifespan as fetal hemoglobin is replaced with adult hemoglobin, and the decreased excretion of bilirubin, secondary to low concentrations of the hepatocyte binding protein, low activity of glucuronosyl transferase, and increased enterohepatic circulation [5]. Neonatal jaundice can be classified as pathological jaundice or physiological jaundice [6]. Physiological jaundice typically appears on days 3–4 of life, requires no intervention, and disappears within 2 weeks. Pathological jaundice is often seen on day 1 of life and worsens over time without intervention.

Some infants carry a higher risk of developing neonatal jaundice than others. Studies have reported prolonged labor, male sex, blood type, and low birthweight causing a higher risk for the development of severe neonatal jaundice [7,8]. Other reported risk factors include previous history of jaundice in the family, primiparity, a history suggestive of inherited red blood cell disorders in the family, including glucose-6-phosphate dehydrogenase (G6PD) deficiency, blood type incompatibility with the mother, birth injuries such as scalp hematoma or significant bruising [7,8], sepsis, hypothermia [9], preterm birth, and exaggerated weight loss post birth [10]. Other studies cite socioeconomic factors related to the mother or household as contributing to the risk of developing severe NNJ [11].

While most cases of neonatal jaundice are self-limiting, some cases are severe and can lead to death or long-term neurodevelopmental disability. NNJ is among the leading causes of hospitalization within the first week post birth [10] and high enough levels of serum bilirubin can result in kernicterus/acute bilirubin encephalopathy. Kernicterus can lead to death, deafness, athetoid cerebral palsy, vision issues, and intellectual disabilities [12]. Pathological jaundice necessitates early detection and treatment to avoid these severe outcomes [10].

Early detection and treatment are hampered in low- and middle-income countries by several factors. First, bilirubin levels often peak on the 4th or 5th day of life, when many infants have already been discharged home from the hospital if born in a facility [12]. In a context of limited postnatal care for newborns, even attentive parents may struggle to notice early skin changes and return to a health facility for clinical management. Second, most facilities lack rapid screening or diagnostic options, such as a transcutaneous bilirubin meter (TcB) that can use refracted light to measure bilirubin levels and has been used to reduce the need for blood draws in newborns [13,14]. Judgements based solely on visual inspection are notoriously challenging [12].

In Ghana, West Africa, screening for jaundice using TcB is not routine in most hospitals, given the cost of a transcutaneous bilirubin meter. Most postnatal wards or clinics detect jaundice in neonates by blanching the skin and/or gums with digital pressure to reveal the underlying color of the skin and subcutaneous tissue. While it is recommended that all newborns be evaluated for jaundice before being discharged, most mothers and babies are sent home in Ghana within 24 h, and only a fraction return for a postnatal visit at 48–72 h old. Others defer postnatal care until their newborn is due for immunizations at 6 weeks, thus decreasing the likelihood of identifying potentially serious jaundice while it is still easily treatable with phototherapy.

Given the profound impact of NNJ in sub-Saharan Africa, we sought to answer the following questions: (1) What percent of admissions within 8 days of birth to the Mother and Baby Unit at our study site were admitted for a primary diagnosis of neonatal jaundice, and what characterized those admissions? (2) How did treatment provided compare to the recommendations of the United Kingdom NICE Guideline CG98 via the “BiliApp”? and (3) Among those with jaundice, what factors were associated with an increased likelihood of severity indicative of the need for blood exchange therapy?

## 2. Materials and Methods

All study methods and reporting were guided by the EQUATOR network’s Strengthening the Reporting of Observational Studies in Epidemiology (STROBE) criteria [15].

### 2.1. Setting

This retrospective chart review study was conducted among all neonates born and admitted to the Suntreso Government Hospital Mother and Baby Unit in Kumasi, Ghana, in 2020. Kumasi is the capital city of Ghana’s Ashanti region, and the nation’s second largest city, with a population of 3.3 million. Suntreso Government Hospital (SGH) is a district hospital that oversees approximately 3000 births per year; its Mother and Baby Unit (MBU) admits sick babies between 0 and 2 months of age. These admissions come from the hospital’s maternity unit, as referrals from other area hospitals, and as self-referrals from home. The MBU has 36 beds and admits approximately 1000 newborns in need of medical care each year. Neonatal jaundice is assessed through visual inspection, followed by serum bilirubin assessment. Serum bilirubin is assessed by taking 2 milliliters of venous blood from the baby’s hand which is then processed by the onsite laboratory, which parents must pay for at the time of service. Serum bilirubin tests costs between 35 and 60 Ghana cedis, or approximately USD 4. Providers carry the sample to the lab for analysis, and total serum bilirubin levels are reported back within 12–24 h. There is no transcutaneous bilirubin meter available for rapid assessment. The unit has single and double phototherapy capabilities, as well as capacity for blood exchange therapy. Suntreso MBU is a level II neonatal unit with two general pediatricians. The MBU has four intern doctors and one pediatric resident who ensure 24 h coverage. Suntreso MBU has a nurse–patient ratio of 1:11.

### 2.2. Data Collection and Analysis

Using the medical record files for neonates admitted to the MBU as a guide, we created a data collection template in Microsoft Excel (Redmond, WA, USA) reflecting key variables of interest. Key maternal variables included maternal age, education, parity, number of antenatal visits attended (and whether they met the World Health Organization recommendation of 8 visits), and delivery type (vaginal vs. cesarean section). Maternal blood type was recorded when available. Key newborn variables included gestational age at birth (typically calculated from last menstrual period or in some cases from ultrasound), birthweight, age at admission (calculated in both hours and in days for ease of interpretation), sex, primary reason for admission, serum bilirubin, treatment with phototherapy, treatment with blood exchange therapy, availability of laboratory bloodwork, blood type when available, G6PD status, feeding method (breastfeeding or formula feeding), length of stay, and outcome (died, discharged, referred). If multiple bilirubin levels were obtained throughout admission, only the bilirubin level at admission was included in our review. Note that “primary reason for admission” is the clinician-reported indication for admission, often recorded before laboratory testing can be conducted and results retrieved, which can take a day or more. In addition, G6PD status is typically assessed for neonates with the most severe jaundice whose mothers can pay for the test.

Three researchers (R.A., B.O., and A.V.W.) collected the files of all neonates admitted to the MBU between 1 January and 31 December 2020, under 8 days of age and entered data. Each file was separately reviewed at least twice. Data were then cleaned and uploaded to Stata 18.0 (College Station, TX, USA), where frequencies, descriptive statistics, and bivariate statistics were calculated. Newborns admitted with a primary diagnosis of jaundice were compared to those admitted for other reasons using Student’s T and Chi square analysis. This comparison was meant to characterize newborns admitted with jaundice relative to other newborns.

Date and time of birth was compared against date and time of admission to calculate ‘hours of life on admission’. Since hospital protocol indicates that newborns with suspected jaundice are to have a serum bilirubin test upon admission, we assumed that serum bilirubin levels in the chart that were recorded on the same day as admission were reflective of bilirubin on admission, unless otherwise indicated. Using gestational age at birth, hours of life at admission, and serum bilirubin levels for neonates admitted for jaundice, we used the BiliApp version 2.0.2 to determine assessment and recommended action for each newborn. The BiliApp is an android and mac-compatible application that is based on the United Kingdom’s NICE Guideline CG98 [16]. After information is entered into the app, the bilirubin level is plotted on the curve representing both the threshold above which phototherapy is recommended and the threshold above which blood exchange therapy is recommended for that gestational age and hour of life. The app then assesses the baby’s serum bilirubin level: normal or below the phototherapy threshold, at or near the phototherapy threshold, above the phototherapy threshold, at or near the exchange threshold, above the exchange threshold. In addition, the app suggests recommended actions: no treatment recommended, repeat assessment in 6–12 h, repeat assessment in 6 h and consider phototherapy, start phototherapy, prepare (initiate) exchange therapy.

After calculating BiliApp recommendations for each newborn with complete data on gestational age, age at admission, and serum bilirubin on admission (*n* = 170), we compared the treatment documented in the chart with the BiliApp recommendations. We also compared the characteristics of those babies who were not treated with those who underwent phototherapy alone and those who underwent phototherapy and exchange therapy. Finally, we conducted multivariate backward stepwise logistic regression to identify, from among those newborns admitted for jaundice, which factors were most strongly associated with a recommended exchange therapy according to the BiliApp. Original variables chosen for the model included those noted in previous literature likely to be associated with differential likelihood of jaundice (maternal parity, delivery type, male sex, breastfeeding, gestational age, birthweight, serum bilirubin level). Maternal and infant blood type and G6PD status were not able to be included due to the limited number of participants with complete data. Non-significant variables were removed one at a time to yield a final model.

All analysis was conducted in Stata 18.0. For all statistical tests, 0.01 was taken for statistical significance.

### 2.3. Ethical Review

This was a secondary analysis of previously collected clinical data spanning the year of 2020, meaning it would not have been feasible to obtain informed consent from every individual whose data were included in the logbooks. Data were instead recorded without any identifiers to ensure privacy. The Committee on Human Research and Publications Ethics at the Kwame Nkrumah University of Science and Technology reviewed and approved the protocol, including waiving the requirement for documented informed consent, and the University of Michigan Institutional Review Board exempted this research from ongoing review given the de-identified nature of the secondary analysis, thus not requiring documentation of informed consent.

## 3. Results

There were 1059 newborns admitted to the MBU in 2020 at less than 8 days of age. One hundred and seventy-nine (16.9%) were admitted with a primary reason for admission listed as neonatal jaundice. Table 1 details the demographics of mothers, who averaged 29 years old and 7.6 antenatal care visits. More than three quarters of mothers had a secondary or tertiary education, and slightly less than a third had just experienced their first birth. Slightly more than half (52.4%) delivered vaginally. When comparing mothers of newborns admitted with jaundice to mothers of newborns admitted with other primary diagnoses, the only significant difference was in the type of delivery. Mothers of jaundiced newborns were more likely to deliver vaginally than mothers of non-jaundiced newborns. (*p* < 0.001)

Table 2 illustrates the characteristics of the newborns, who averaged 38.3 weeks gestation at birth. Of all newborns admitted to the unit, more than half (56.6%) were male, 19.6% were preterm, and 22.7% were considered low birthweight (<2500 g). Birth asphyxia (17.3%) and neonatal jaundice (16.9%) were the most common reasons for admission. Slightly more than half (53.9%) had serum bilirubin assessed at the time of admission, and 10% had other bloodwork information in their chart. The most commonly reported outcome of laboratory test related to hemoglobin levels, which ranged from 5.8 to 23.0 g/dL. Additionally, 23.9% (*n* = 253) had G6PD status assessed. Of those, 1 had a partial defect and 36 had a full defect (14.6% of those assessed). The vast majority of newborns were breastfed (*n* = 968, 95.9%) and were admitted within the first 24 h (*n* = 682, 64.4%). Thirty-nine babies (3.8%) died, 45 (4.3%) were referred, and 956 (91.1%) were discharged. The mean length of stay was 4.9 days.

Table 2 also illustrates the characteristics of those newborns admitted for jaundice, compared with newborns admitted in the first week after birth for other reasons. Differences noted include age at admission—with jaundiced babies admitted at an average of 3.6 days after birth, compared to babies admitted for other causes being admitted at an average of less than half a day after birth (0.44, *p* = 0.001). Jaundiced newborns had a shorter length of stay (3.6 days) than newborns admitted for other reasons (5.1 days, *p* < 0.001). Jaundiced newborns were less likely than non-jaundiced babies to be born before 37 weeks gestation (10.9% vs. 21.3%, *p* = 0.006), and less likely to be low birthweight (8.9% vs. 25.5%, *p* < 0.001). Additional differences include expected factors related to jaundice, with a higher percentage of jaundiced babies having/receiving any diagnosis of jaundice (not just primary), serum bilirubin levels, treatment with phototherapy, and treatment with exchange therapy, and G6PD status (all significant at *p* < 0.001). Only 53.1% of newborns admitted with a primary diagnosis of jaundice had their G6PD status assessed (*n* = 95), and among those, 17 had a full defect and 1 had a partial defect. There was no difference between groups in terms of likelihood of being breastfed. In terms of outcomes, jaundiced babies were more likely to be referred to higher level facilities than newborns admitted for other reasons.

Figure 1 illustrates the distribution of serum bilirubin levels by hours of life at admission for newborns with a primary diagnosis of jaundice, indicating a wide range of both bilirubin levels and hours of life at admission. Table 3 reflects the BiliApp assessment for the newborns admitted with a primary diagnosis of jaundice in addition to comparing BiliApp-recommended treatment with actual treatment received. A total of 170 out of 179 had complete data for gestational age at birth, serum bilirubin, and date and time of birth and admission, allowing for assessment to be completed using the BiliApp. The BiliApp recommended no treatment for 36 neonates (21.2% of newborns admitted for jaundice). Of those 36, 26 were treated with phototherapy alone and 1 was treated with phototherapy and exchange therapy. At the other end of the spectrum, the BiliApp recommended exchange therapy for 41 newborns who were admitted for jaundice, 21 of whom received it. Three newborns received exchange therapy who were not recommended to receive it by the BiliApp.

Table 4 illustrates the differences between neonates admitted for jaundice based on the treatment received (no treatment (*n* = 14), phototherapy only (*n* = 137), phototherapy and exchange therapy (*n* = 24)). The only significant differences across groups were gestational age, with those receiving exchange therapy being slightly younger (*p* = 0.05); bilirubin levels on admission, with those receiving exchange therapy having higher levels (*p* < 0.001); breastfeeding, with those receiving exchange therapy having lower rates of breastfeeding (*p* = 0.04); and outcome, with those receiving exchange therapy being much more likely to die or be referred than those who did not receive exchange therapy (*p* < 0.001).

Finally, Table 5 illustrates the results of the backward stepwise multivariate logistic regression to identify factors associated with being recommended for exchange therapy. Although key risk factors of G6PD status and parent/newborn blood group incompatibility were not able to be included in the model due to small sample sizes, the model suggests serum bilirubin and gestational age are the most significant factors, even after controlling for exclusive breastfeeding and male sex. Note the pseudo R^2^ of the model is 0.48, indicating that this model accounts for approximately half of the variability in the risk of being recommended for exchange therapy.

## 4. Discussion

Approximately one out of every six babies admitted to the Mother and Baby Unit at our study hospital in 2020 were admitted with a primary diagnosis of neonatal jaundice. Treatment was not always consistent with retrospectively assessed recommendations from the BiliApp, an application that calculates severity of jaundice based on gestational age at birth, hours of age at assessment, and serum bilirubin levels. According to the BiliApp, 29.4% (*n* = 50) of newborns’ bilirubin was normal, and no treatment or reassessment was recommended for 36 of the 50. Of those 36 newborns with “normal” bilirubin low enough to not require further action, 9 were given no treatment, 26 were given phototherapy, and one was given both phototherapy and exchange therapy. The BiliApp also recommended exchange therapy for 24.1% of newborns (*n* = 41). Of those 41, 20 received phototherapy and 21 were given both phototherapy and exchange therapy.

This discordance raises important questions about the diagnosis and treatment of neonatal jaundice in low-resource settings. First, the hospital where this study was conducted does not have a transcutaneous bilirubin meter, and thus they are unable to offer rapid, bedside assessment for newborns with suspected jaundice. Instead, providers must examine the newborn and use their judgement as to whether the newborn is jaundiced or not. If jaundice is suspected, and if the mother has the money and agrees to pay for a test, a blood test is ordered. At the time this study was conducted, laboratory results took between 12 and 24 h to return, depending on the day of the week and the timing of the test. Thus, providers were put in the position of waiting for the blood work to come back, and potentially allowing jaundice to progress to more dangerous levels, or presumptively treating with phototherapy. It appears that providers typically opted to presumptively treat with phototherapy, based upon the number of newborns who received phototherapy compared to the number whose BiliApp scores suggested they did not need it. At the same time, fewer newborns received exchange therapy than was indicated by the BiliApp. It is not surprising that providers underutilized this procedure, which is typically reserved for the most severely ill and can be very dangerous to execute. Nonetheless, it warrants further research to determine how outcomes may have been affected by these treatment patterns. It is also worth noting that the ratio of nurses to babies (1:11) in the unit may contribute to discordance between treatment given and United Kingdom NICE Guideline CG98.

The scatterplot in Figure 1 illustrates that newborns being admitted for jaundice range from those identified within hours of birth to those who are nearly a week old, suggesting that the window for identifying jaundice is quite wide in this setting. Screening is necessary throughout the first week, not simply at discharge and at the recommended 1-week postnatal visit. Similarly, the serum bilirubin levels are extremely variable, clustering below 300 for the youngest newborns and ranging from below 100 to more than 600 for babies older than 2 days old. This suggests that there is variability in recognition of jaundice, which is not surprising given the reliance on visual examination as a screening mechanism.

While the risks of presumptive treatment with phototherapy may be outweighed by the risk of delaying the initiation of treatment for a newborn suspected of having jaundice, phototherapy is not a benign intervention [17]. Short-term side effects include potential interference with maternal–newborn bonding, circadian cycle disruption, skin lesions, and damage to the eyes [17]. Longer-term side effects include potential links to childhood cancers and immune system dysregulation [17]. Thus, presumptive treatment—while understandable—is not ideal.

Narayanan et al. (2019) describe the challenges for health facilities in LMICs to treat ill newborns with limited technology and resources [18]. Indeed, the recorded diagnoses supporting admission to the MBU in our study were typically based on clinical judgement, rather than confirmed laboratory testing. The percentage of newborns with jaundice as an admitting diagnosis in our study (16.9%) fell below the values reported by previous similar studies: 17.9% in Nigeria [19], 26.5% in Benin [20], and 32.9% elsewhere in Ghana [21]. However, the total percentage of admitted babies with jaundice at our study site was higher (56.5%), perhaps pointing to discrepancies in classification of jaundice as the primary medical problem.

In this study, we found expected differences between newborns admitted to the MBU for neonatal jaundice when compared to those being admitted for other issues. Jaundiced babies were older at birth and older at admission that non-jaundiced babies, and they were more likely to have undergone phototherapy, exchange therapy, and have had bloodwork completed. Notably, the completion of bloodwork was uncommon in this study population, with only 10% of all newborns having the results of bloodwork documented in their charts. These findings are a reminder of the challenges associated with “pay as you go” healthcare, in which patients and families are expected to pay for laboratory tests as they are needed. Even though Ghana has a National Health Insurance Scheme (NHIS) that theoretically covers the cost of maternal and newborn healthcare, data suggest that out-of-pocket payments remain common and serve as a barrier to care [22,23]. Anecdotally, our team routinely sees families paying for blood tests and serum bilirubin tests, regardless of whether they should be covered by NHIS.

Our study also examined the factors associated with receiving a BiliApp recommendation for blood exchange therapy. Given the level of missing data that precluded the inclusion of expected correlates such as G6PD status or ABO incompatibility, the model indicated that only gestational age and serum bilirubin levels were predictive of an exchange recommendation, even after controlling for breastfeeding and male sex. The model’s pseudo R^2^ indicates that the model explains nearly half of the variance in the prediction of exchange therapy, yet the predictors are not particularly surprising or novel. Higher serum bilirubin is linked to a moderate increase in risk of being recommended for exchange therapy, and older gestational age is linked to a 51% decrease in the risk. This may be due to the increased risk of bilirubin toxicity among the youngest newborns, with the risk of toxicity dropping as gestational age increases. Other potential correlates, such as maternal age, maternal education, and parity, dropped out of the model due to statistical insignificance. Notably, the model that included these non-significant variables had a pseudo R^2^ only 1 percent higher than our final model.

This study has several limitations worth noting. First, data were limited to that which are routinely collected and documented at the time of admission. Missing data from paper charts was a challenge, and despite working hard to track down values for every variable, our dataset did suffer from inconsistent Ns related to missing data. With the exception of maternal education and newborn feeding type, both of which had high levels of missing data, missingness hovered around 1% of total values. In addition, our dataset did not include such factors as maternal socioeconomic status, nor did it allow for longitudinal follow-up to assess neurologic status or hearing that might have been affected by bilirubin toxicity. We also used the first serum bilirubin level reported with the assumption that blood would have been drawn near the time of admission for a baby suspected of having jaundice, allowing us to determine the timing of bilirubin assessment based on the timing of admission. However, it is possible that serum bilirubin testing did not occur immediately after admission. It is also possible that neonates’ bilirubin levels assessed at admission were not reflective of serum bilirubin levels—or visual inspection or clinical indications—later in their stay, which could explain discrepancies between BiliApp recommendations and actual treatment patterns. We also did not have access to data regarding providers’ visual inspection to compare treatment against provider judgment. Another limitation of this study is that data were from 2020, the first year of the COVID-19 pandemic. While the impact of COVID-19 was significantly less in Ghana than many other places (with fewer than 55,000 cases identified by the end of 2020, out of a country of nearly 30 million people), it is possible that data from 2020 may not reflect the reality of non-COVID-19 years [24]. Finally, the level of missing data regarding serum bilirubin introduces the possibility of bias, given that less than 54% of neonates admitted to the MBU had serum bilirubin assessed at the time of admission. Future research that includes payment for SBR for all study participants is warranted.

The strengths of our study include its comprehensive study population, including a full year of data from a busy referral hospital. This study also relied upon written records that were reviewed twice by trained research staff, minimizing the risk of recall bias. This study also utilized standardized United Kingdom treatment guidelines to determine recommended treatments for jaundice.

Regardless, this study has several important implications. Our findings suggest that the lack of rapid bedside diagnostics in many low-resource settings can lead to presumptive treatment for jaundice, well before providers have confirmation of a newborn’s bilirubin level. This leads to overtreatment for some newborns, typically by starting phototherapy when observation and repeated testing might be better aligned with guidelines. At the same time, our study indicates a high percentage of newborns are admitted with bilirubin levels that are high enough to warrant blood exchange therapy, even though only about half of such newborns received exchange therapy. Prioritizing the purchase of transcutaneous bilirubin meters (TcBs) when possible and developing lower-cost alternatives will be an important step in more rapidly triaging newborns and improving the management of neonatal jaundice in low-resource settings. It has been established that transcutaneous bilirubin assessments are not perfect—they may overestimate serum bilirubin among dark-skinned infants and lead to overtreatment if not compared with serum bilirubin [25,26]. Thus, future work should prioritize the development and implementation of novel assessment mechanisms, including artificial intelligence-based algorithms and smart phone-based applications [27,28]. On a smaller scale, quality improvement initiatives that seek to improve screening and data capture, educate families about the signs of jaundice, and align clinical care with international guidelines are warranted.

In addition, further research that explores neonatal jaundice screening outside the hospital setting is warranted. Our study also highlighted gaps in the number of newborns who receive blood tests, which is likely attributable to the expectation that in Ghana, tests are paid for in advance. Including such tests in Ghana’s National Health Insurance Scheme, and ensuring they are indeed covered, is an important step to improving newborn outcomes.

In summary, this study highlights significant challenges in diagnosing and treating neonatal jaundice in low-resource settings, as demonstrated at one referral hospital in urban Ghana. A considerable proportion of newborns were treated for jaundice without confirmation from rapid, reliable diagnostics, largely due to the absence of transcutaneous bilirubin meters. This lack of equipment often led healthcare providers to presumptively treat newborns with phototherapy rather than waiting for laboratory results, which were delayed due to logistical and financial constraints. Although this approach may mitigate the risk of delaying necessary treatment, it also raises concerns about overtreatment and its potential long-term effects on newborns. Furthermore, there was a notable discrepancy between recommended and actual treatments, with only about half of the infants who required exchange therapy receiving it.

These findings underscore the critical need for improved diagnostic capabilities, such as affordable transcutaneous bilirubin meters, which would allow for more accurate and timely assessments of infants at risk for jaundice. Additionally, there is an urgent call to integrate necessary laboratory tests into Ghana’s National Health Insurance Scheme to alleviate the financial burden on families and improve care accessibility and adherence to recommended treatment guidelines. By addressing these gaps, we can enhance the management of neonatal jaundice and potentially improve newborn health outcomes in low-resource settings. Further research into neonatal jaundice screening and treatment practices outside the hospital setting could also provide valuable insights for improving care strategies.

## Figures and Tables

**Figure 1 ijerph-22-01032-f001:**
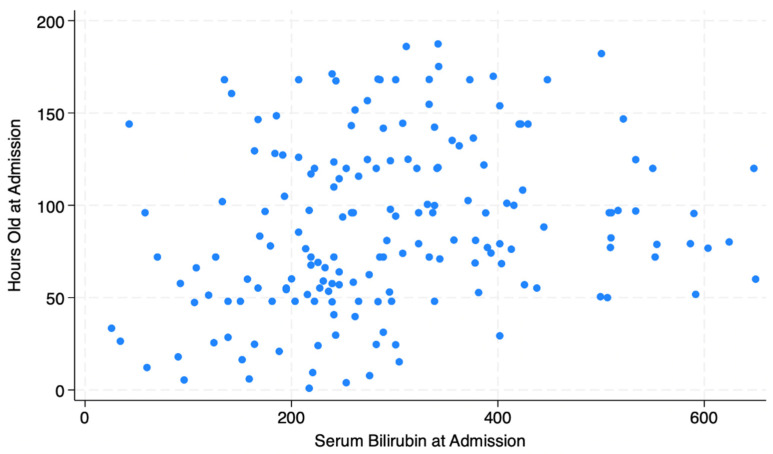
Scatterplot of serum bilirubin levels on admission against age at admission.

**Table 1 ijerph-22-01032-t001:** Overall characteristics of mothers in our sample (*n* = 1059) *.

	Overall Sample (*n* = 1059)	Sample Admitted for Jaundice (*n* = 179)	Sample Admitted for Other Primary Concerns (*n* = 880)	*p* Value
Continuous Variables	Mean (95% CI)			
Maternal Age (years) (*n* = 1050)	29.1 (28.7–29.4)	29.6 (28.8–30.5)	29.0 (28.6–29.4)	0.200
Number of antenatal care visits	7.6 (7.4–7.7)	7.7 (7.4–8.1)	7.5 (7.4–7.7)	0.374
Categorical Variables	*n* (%)			
Maternal Age (years)				0.166
5–19	63 (5.9)	12 (6.7)	51 (5.8)	
20–29	486 (45.9)	69 (38.6)	417 (47.4)	
30–39	461 (43.5)	87 (48.6)	374 (42.5)	
40+	49 (4.6)	11 (6.2)	38 (4.3)	
Maternal Education	(*n* = 923) 136 missing	(*n* = 164) 15 missing	(*n* = 759) 121 missing	0.120
None	48 (5.2)	9 (5.5)	39 (5.1)	
Primary	64 (6.9)	8 (4.9)	56 (7.4)	
Secondary	602 (65.2)	99 (60.4)	503 (66.3)	
Tertiary	209 (22.6)	48 (29.3)	161 (21.2)	
Parity	(*n* = 1050) 9 missing	(*n* = 175) 4 missing	(*n* = 875) 5 missing	0.353
1	337 (32.1)	41 (24.0)	297 (33.9)	
2	276 (26.3)	59 (33.9)	214 (24.5)	
3	191 (18.1)	29 (16.6)	162 (18.5)	
4	126 (12.0)	28 (16.1)	99 (11.3)	
5+	120 (10.5)	17 (9.7)	103 (11.8)	
Antenatal care				0.498
Received recommended 8+ visits	591 (55.8)	104 (58.1)	487 (55.3)	
Delivery type	(*n* = 1055) 4 missing	(*n* = 177) 2 missing	(*n* = 878) 2 missing	<0.001
Vaginal	553 (52.4)	117 (66.1)	436 (49.7)	
Cesarean	502 (47.6)	60 (33.9)	442 (50.3)	

* When missing values reduced the sample size, the adjusted sample size is noted.

**Table 2 ijerph-22-01032-t002:** Overall characteristics of newborns in our sample (*n* = 1059).

	Overall Sample (*n* = 1059)	Sample Admitted for Jaundice (*n* = 179)	Sample Admitted for Other Primary Concerns (*n* = 880)	*p* Value
Continuous Variables	Mean (95% CI)			
Gestational age at birth (wks) (*n* = 1048, 11 missing)	38.3 (38.2–38.5)	38.8 (38.5–39.0)	38.3 (38.0–38.5)	0.042
Birthweight (in grams) (*n* = 1055, 4 missing)	2963 (2914–3011)	3085 (3012–3159)	2939 (2886–2995)	0.026
Age at admission (in days) (*n* = 1058, 1 missing)	0.97 (0.87–1.07)	3.6 (3.3–3.9)	0.44 (0.37–0.50)	0.001
Length of stay (in days)	4.9 (4.6–5.1)	3.5 (2.9–3.1)	5.1 (4.2–4.9)	<0.001
Mean Serum Bilirubin	197.4 (188.1–206.7)	296.1 (276.4–315.9)	153.9 (147.3–160.6)	<0.001
Categorical Variables	*n* (%)			
Baby Sex	(*n* = 1057) 2 missing			0.651
Male	598 (56.6)	104 (58.1)	494 (56.3)	
Female	459 (43.4)	75 (41.9)	384 (43.7)	
Gestational age	(*n* = 1048) 11 missing	(*n* = 174) 5 missing	(*n* = 874) 6 missing	0.006
Preterm (<37 weeks)	205 (19.6)	19 (10.9)	186 (21.3)	
Term (37–42 weeks)	811 (77.4)	150 (86.2)	661 (75.6)	
Post-dates (>42 weeks)	32 (3.1)	5 (2.9)	27 (3.1)	
Birthweight				<0.001
Low birthweight (<2500 g)	240 (22.7)	16 (8.9)	224 (25.5)	
Normal (2500–4000 g)	729 (68.8)	158 (88.3)	571 (64.9)	
Macrosomia (>4000 g)	90 (8.5)	5 (2.8)	85 (9.7)	
Primary reason for admission *				not analyzed due to small sample sizes
Fetal anomaly	20 (1.9)	0 (0)	20 (2.3)	
Birth asphyxia	183 (17.3)	1 (0.6)	182 (20.7)	
Hypothermia	96 (9.1)	0 (0)	96 (10.9)	
Fever	95 (9.0)	12 (6.7)	83 (9.4)	
Infection	25 (2.4)	5 (2.8)	20 (2.3)	
Jaundice	179 (16.9)	179 (100.0)	0 (0)	
Low birthweight	106 (10.0)	1 (0.6)	105 (11.9)	
Macrosomia	80 (7.5)	0 (0)	80 (9.1)	
Meconium-stained liquor	94 (8.9)	0 (0)	94 (10.7)	
Prematurity	107(10.1)	0 (0)	107 (12.2)	
Respiratory issues	116 (11.0)	2 (1.1)	114 (13.0)	
Sepsis	57 (5.4)	8 (4.5)	49 (5.6)	
Other	35 (3.3)	2 (1.1)	33 (3.8)	
Any diagnosis of jaundice	598 (56.5)	179 (100.0)	418 (47.6)	<0.001
Treated with phototherapy	406 (38.4)	165 (92.2)	241 (27.4)	<0.001
Treated with exchange therapy	27 (2.6)	26 (14.6)	1 (0.1)	<0.001
Bloodwork completed	106 (10.0)	51 (28.5)	55 (6.3)	<0.001
Serum Bilirubin assessed	571 (53.9)	175 (97.8)	396 (45.0)	<0.001
G6PD status assessed	253 (23.9)	95 (53.1)	158 (18.0)	<0.001
G6PD status (*n* = 253)		(*n* = 95)	(*n* = 158)	<0.001
No defect	216 (85.4)	77 (81.1)	139 (87.3)	
Partial defect	1 (0.4)	1 (1.1)	0 (0)	
Full defect	36 (14.2)	17 (17.9)	19 (12.1)	
Type of feeding (*n* = 1009) 50 missing	(*n* = 1027) 50 missing	(*n* = 171) 8 missing	(*n* = 838) 42 missing	0.655
Breastfeeding	968 (95.9)	163 (95.3)	805 (96.1)	
Formula feeding	41 (4.1)	8 (4.7)	33 (3.9)	
Age at admission (hours)	(*n* = 1058) 1 missing	(*n* = 178) 1 missing		<0.001
<24 h	689 (65.1)	11 (6.2)	678 (77.0)	
24–48 h	179 (16.9)	25 (14.0)	154 (17.5)	
49–76 h	74 (7.0)	44 (24.7)	30 (3.4)	
>76 h	116 (10.9)	98 (55.1)	18 (2.0)	
Outcome	(*n* = 1039) 20 missing	(*n* = 175) 4 missing	(*n* = 864) 16 missing	<0.001
Died	39 (3.8)	7 (4.0)	32 (3.7)	
Referred	45 (4.3)	17 (9.7)	28 (3.2)	
Discharged	955 (91.9)	151 (86.3)	804 (93.1)	

* Totals exceed sample size for reason for admission because two reasons were often cited together as the primary reason for admission (e.g., low birthweight/prematurity).

**Table 3 ijerph-22-01032-t003:** Clinical assessment and recommendation for newborns admitted with jaundice as the primary reason (with complete data on gestational age, hours of life at admission, and serum bilirubin level at admission) according to the BiliApp * (*n* = 170).

BiliApp Assessment	Overall *n* (%)			
Normal/below phototherapy threshold	50 (29.4)			
At or near phototherapy threshold	43 (25.3)			
Above phototherapy threshold	36 (21.2)			
Above exchange threshold	41 (24.1)			
BiliApp Recommendation	(*n* = 170)	Actual treatment received		
	(Overall)	No treatment given (*n* = 13)	Phototherapy Only (*n* = 133)	Phototherapy and Exchange Therapy Given (*n* = 24)
No treatment recommended	36 (21.2)	9 (69.2)	26 (19.5)	1 (4.2)
Repeat assessment in 6–12 h	21 (12.4)	3 (23.1)	18 (13.5)	0
Repeat assessment in 6 h, consider phototherapy	36 (21.2)	1 (7.7)	35 (26.3)	0
Start phototherapy	36 (21.2)	0	34 (25.6)	2 (8.3)
Prepare exchange therapy	41 (24.1)	0	20 (15.1)	21 (87.5)

* Based on the United Kingdom’s NICE Guideline CG98 [16].

**Table 4 ijerph-22-01032-t004:** Characteristics of neonates admitted for jaundice based on the treatment received (*n* = 175).

	No Treatment Given (*n* = 14) (Mean, 95%CI)	Phototherapy Only (*n* = 137) (Mean, 95%CI)	Phototherapy and Exchange Therapy Given (*n* = 24) (Mean, 95%CI)	*p* Value *
Maternal age (yrs)	28.2 (25.5, 30.6)	29.7 (28.8, 30.7)	29.3 (26.9, 31.7)	0.59
Maternal parity	2.3 (1.8, 2.8)	2.6 (2.4, 2.9)	2.7 (1.9, 3.4)	0.72
Number of antenatal care visits	6.5 (5.1, 7.9)	7.9 (7.4, 8.3)	7.9 (6.8, 8.9)	0.22
Gestational age (wks) (*n* = 173)	38.8 (37.9, 39.5)	38.9 (38.7, 39.2)	38.1 (37.4, 38.7)	0.05 **
Birthweight (g)	3178 (2969, 3388)	3098 (3013, 3184)	2900 (2717, 3084)	0.13
Hours of age at admission	93.6 (63.9, 123.4)	90.8 (83.4, 98.2)	72.9 (55.9, 90.0)	0.18
Bilirubin level on admission (*n* = 174)	142.3 (98.1, 186.5)	288.3 (269.3, 307.3)	434.8 (379.6, 490.1)	<0.001 **
	*n* (%)	*n* (%)	*n* (%)	
Breastfeeding	12 (92.3)	131 (94.2)	20 (83.3)	0.04 **
Male sex	7 (53.8)	81 (60.9)	15 (62.5)	0.81
Outcome				<0.001 **
Died	0 (0)	5 (3.7)	2 (8.3)	
Referred	1 (7.1)	5 (3.7)	11(45.8)	
Discharged	13 (92.9)	127 (92.7)	11 (45.8)	

* means tested via ANOVA, F statistic; categorical variables tested using Chi^2^; ** *p* ≤ 0.05.

**Table 5 ijerph-22-01032-t005:** Multivariate regression analysis of predictors of a BiliApp recommendation to start exchange therapy *.

Variable	OR (95% CI)	Standard Error	Z	*p* Value
Serum bilirubin	1.01 (1.01–1.02)	0.003	5.84	<0.001
Exclusive breastfeeding	0.32 (0.05–2.06)	0.31	−1.18	0.236
Male newborn	0.95 (0.34–2.66)	0.49	−0.08	0.936
Gestational age	0.51 (0.35–0.74)	0.10	−3.59	<0.001

* Log likelihood = −48.9; LR chi^2^ = 90.6; prob > chi^2^ = <0.0001; Pseudo R^2^ = 0.481.

## Data Availability

The data/dataset used in this current study are available from the corresponding author upon reasonable request.

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
