# Peer review of "Neonatal Jaundice Treatment Versus Recommendations: The Challenge of Treatment Without Rapid Diagnostic Capability"

_ijerph, 2025, doi:10.3390/ijerph22071032_

Round 1

Reviewer 1 Report

Comments and Suggestions for Authors

General Comments

The retrospective review of records is well done, thoughtfully analyzed, and provides valuable conclusions that could encourage the wider utilization of the transcutaneous bilirubin meter.

Introduction

Last paragraph, question 1 says, “What percent of admissions within 8 days of birth…” This contradicts question 1 in the abstract’s “Methods” section, “What percent of admissions within 7 days of birth…”

Materials and Methods

Last sentence of the paragraph labeled “Setting” mentions a nurse:patient ratio in the Suntreso MBU hospital of 11:1, which is appalling, unsafe, and suggests that patients are likely not treated in a timely manner.  This statistic may in some way affect results and could warrant a comment in the discussion section.

Results

Last paragraph reports that less than 54% of neonates admitted to MBU had serum bilirubin assessed at the time of admissions.  This hole in the data introduces a bias in the results that should be dealt with in the discussion.

Discussion

In general, all data worth collecting is worth at least a passing comment in the discussion, even if treated as part of a group.  Can you comment on the Figure 1 scatterplot in the Results section?  What is the significance of the fact that you found a “wide range of both bilirubin levels and hours of life at admission”?

Fifth paragraph:  Can you speculate on why parents seemed to be paying out-of-pocket for lab tests that you say are covered by government health insurance?  Is it because it speeds up the service?

Sixth paragraph, fourth sentence:  “Higher serum bilirubin is linked to a moderate increase in risk of being recommended for exchange therapy, and older gestational age is linked to 51% decrease in the risk.”  This is a summary of results, but does not comment.  Can you provide a hypothesis for at least the interesting and counterintuitive second phrase of that sentence?

Thank you, and I enjoyed reading your paper.  Strong work.

Author Response

R1, C1: The retrospective review of records is well done, thoughtfully analyzed, and provides valuable conclusions that could encourage the wider utilization of the transcutaneous bilirubin meter.

Response: Thank you for your careful review of our manuscript.

R1, C2: Last paragraph, question 1 says, “What percent of admissions within 8 days of birth…” This contradicts question 1 in the abstract’s “Methods” section, “What percent of admissions within 7 days of birth…”

Response: Thank for your spotting this inconsistency! The abstract has been updated to align with the rest of the manuscript, which states “within 8 days of birth”.

R1, C3: Last sentence of the paragraph labeled “Setting” mentions a nurse:patient ratio in the Suntreso MBU hospital of 11:1, which is appalling, unsafe, and suggests that patients are likely not treated in a timely manner.  This statistic may in some way affect results and could warrant a comment in the discussion section.

Response: While we agree that that nurse:patient ratio is extremely low, unfortunately, this is not uncommon in low-resource settings such as Ghana. We have added a line to the discussion regarding this issue:  “It is also worth noting that the ratio of nurses to babies (1:11) in the unit may contribute to discordance between treatment given and United Kingdom NICE Guideline CG98.”

R1, C4: Last paragraph reports that less than 54% of neonates admitted to MBU had serum bilirubin assessed at the time of admissions.  This hole in the data introduces a bias in the results that should be dealt with in the discussion.

Response: Thank you for this. We have added the following to the discussion: “Finally, the level of missing data regarding serum bilirubin introduces the possibility of bias, given that less than 54% of neonates admitted to the MBU had serum bilirubin assessed at the time of admission. Future research that includes payment for SBR for all study participants is warranted.”

R1, C5: In general, all data worth collecting is worth at least a passing comment in the discussion, even if treated as part of a group.  Can you comment on the Figure 1 scatterplot in the Results section?  What is the significance of the fact that you found a “wide range of both bilirubin levels and hours of life at admission”?

Response: Thank you for this comment. We have added the following text to the discussion.

“The scatterplot in Figure 1 illustrates that newborns being admitted for jaundice range from those identified within hours of birth to those who are nearly a week old, suggesting that the window for identifying jaundice is quite wide in this setting. Screening is necessary throughout the first week, not simply at discharge and at the recommended 1-week postnatal visit. Similarly, the serum bilirubin levels are extremely variable, clustering below 300 for the youngest newborns and ranging from below 100 to more than 600 for babies older than 2 days old. This suggests that there is variability in recognition of jaundice, which is not surprising given the reliance on visual examination as a screening mechanism.”

R1, C6: Fifth paragraph:  Can you speculate on why parents seemed to be paying out-of-pocket for lab tests that you say are covered by government health insurance?  Is it because it speeds up the service?

Response: This is an excellent question that unfortunately has a complicated answer. Our previous work indicates that government reimbursements are slow, may be less than the true cost, may be limited to certain items that may or may not be available, may not cover ‘off site’ laboratory services when on-site labs experience stock-outs, and hospitals charge patients to be able to keep the facility running. This may or may not be legal – as hospitals should not be charging for things they will eventually be reimbursed for – but there does not seem to be significant concern / political will to fight this despite lots of research and documentation of the challenges of out-of-pocket payments in a system that is supposed to cover all maternal and newborn care. Without getting into the details or accusing the government of wrongdoing, we cited our previous work (refs 22 and 23) that describes the issues.

R1, C7: Sixth paragraph, fourth sentence:  “Higher serum bilirubin is linked to a moderate increase in risk of being recommended for exchange therapy, and older gestational age is linked to 51% decrease in the risk.”  This is a summary of results, but does not comment.  Can you provide a hypothesis for at least the interesting and counterintuitive second phrase of that sentence?

Thank you, and I enjoyed reading your paper.  Strong work.

Response: Thank you for this comment. The following was added to the discussion: “Higher serum bilirubin is linked to a moderate increase in risk of being recommended for exchange therapy, and older gestational age is linked to 51% decrease in the risk. This may be due to the increased risk of bilirubin toxicity among the youngest newborns, with the risk of toxicity dropping as gestational age increases.”

Reviewer 2 Report

Comments and Suggestions for Authors

-The main question raised by Drs Bakari et al is whether treatment thresholds suggested by UK recommendations are followed by local staff in Kumasi, Ghana. To answer this question the authors undertook a retrospective audit examining infants presenting to the maternity service. The characteristics of infants presenting with a diagnosis of hyperbilirubinaemia were then used to assess type of treatment received and compare this to an app (BiliAPP, NICE, UK).

The authors raise an important question as the global burden of neonatal hyperbilirubinaemia is significant and is highest in low and middle income countries (Olusanya et al ., Pediatrics 2018 Feb;141(2):e20171471.). Data from this region in Ghana were not previously available and therefore constitute an important set of data to inform local policy-making.

Introduction:

-The introduction is of sufficient detail for the reader to understand the pathophysiology of neonatal hyperbilirubinaemia, to understand the local context, in particular how infants are managed and what the barriers to access to care and testing are.

Methods:

The methodological approach is detailed and explains the process of data extraction, with emphasis on collecting variables that are associated with the development of hyperbilirubiaemia. The authors then performed stepwise logistic regression to assess which factors were associated with an exchange transfusion. Variables that were too infrequently collected (G6PD status and blood group incompatibility) were omitted. The statistical analysis is appropriate.

Results:

The results are presented in a clear format. The results are explained in the accompanying text.

Discussion:

The authors discuss the strengths and limitations adequately. In particular, the time lapse between obtaining the blood sample and the treatment initiation could introduce unmeasured effects in that the disease has progressed significantly (12-24h delay in obtaining laboratory results) It is possible that clinical features of bilirubin-induced neurological dysfunction had evolved while laboratory results were being awaited.

The authors also acknowledge the lack of follow-up to assess neurological status. The authors could include in the discussion that hearing assessments could be included in the follow-up of high-risk infants.

Bilirubin is difficult to measure transcutaneously above higher thresholds and serum bilirubin would still to be measured. Transcutaneous bilirubinometry overestimates levels as well and could result in overtreatment with its possible side effects (Olusanya BO, Imosemi DO, Emokpae AA. Differences between transcutaneous and serum bilirubin measurements in black African neonates. Pediatrics 2016; 138: e2016090).

Novel approaches to assessing jaundiced newborn could also be discussed such e.g. artificial intelligence-based algorithms and phone-based applications (Saloojee H S Afr Fam Pract (2004). 2024 Feb 29;66(1):5833).

The authors could formulate a quality improvement initiative to try and integrate the gaps identified in e.g. patient documentation and pre-discharge education for parents.

The references are appropriate and relevant to the topic discussed.

Style suggestions:

L66: -micromol / l or μmol/l as units

L389:-suggest: ..include aspects (or factors) such as …

Thank you.

Author Response

R2, C1: The main question raised by Drs Bakari et al is whether treatment thresholds suggested by UK recommendations are followed by local staff in Kumasi, Ghana. To answer this question the authors undertook a retrospective audit examining infants presenting to the maternity service. The characteristics of infants presenting with a diagnosis of hyperbilirubinaemia were then used to assess type of treatment received and compare this to an app (BiliAPP, NICE, UK). The authors raise an important question as the global burden of neonatal hyperbilirubinaemia is significant and is highest in low and middle income countries (Olusanya et al ., Pediatrics 2018 Feb;141(2):e20171471.). Data from this region in Ghana were not previously available and therefore constitute an important set of data to inform local policy-making.

Response: Thank you for this comment, and we appreciate your careful review of this study.

R2, C2: The introduction is of sufficient detail for the reader to understand the pathophysiology of neonatal hyperbilirubinaemia, to understand the local context, in particular how infants are managed and what the barriers to access to care and testing are.

Response: Thank you.

R2, C3: The methodological approach is detailed and explains the process of data extraction, with emphasis on collecting variables that are associated with the development of hyperbilirubiaemia. The authors then performed stepwise logistic regression to assess which factors were associated with an exchange transfusion. Variables that were too infrequently collected (G6PD status and blood group incompatibility) were omitted. The statistical analysis is appropriate.

Response: Thank you again for your careful review of our manuscript.

R2, C4: The results are presented in a clear format. The results are explained in the accompanying text.

Response: Thank you.

R2, C5: The authors discuss the strengths and limitations adequately. In particular, the time lapse between obtaining the blood sample and the treatment initiation could introduce unmeasured effects in that the disease has progressed significantly (12-24h delay in obtaining laboratory results) It is possible that clinical features of bilirubin-induced neurological dysfunction had evolved while laboratory results were being awaited. The authors also acknowledge the lack of follow-up to assess neurological status. The authors could include in the discussion that hearing assessments could be included in the follow-up of high-risk infants.

Response: Thank you for this comment. We have added a statement about follow-up of high-risk infants in the discussion.

R2, C6: Bilirubin is difficult to measure transcutaneously above higher thresholds and serum bilirubin would still (need) to be measured. Transcutaneous bilirubinometry overestimates levels as well and could result in overtreatment with its possible side effects (Olusanya BO, Imosemi DO, Emokpae AA. Differences between transcutaneous and serum bilirubin measurements in black African neonates. Pediatrics 2016; 138: e2016090).

Novel approaches to assessing jaundiced newborn could also be discussed such e.g. artificial intelligence-based algorithms and phone-based applications (Saloojee H S Afr Fam Pract (2004). 2024 Feb 29;66(1):5833).

Response: Thank you for this comment. We have added the following text to the discussion to address both the limitations of transcutaneous measurement and novel approaches to assessing jaundice: “It has been established that transcutaneous bilirubin assessments are not perfect – they may overestimate serum bilirubin among dark-skinned infants and lead to overtreatment if not compared with serum bilirubin.[25,26] Thus future work should prioritize the development and implementation of novel assessment mechanisms, including artificial intelligence-based algorithms and smart phone-based applications.[27,28]”

R2, C7: The authors could formulate a quality improvement initiative to try and integrate the gaps identified in e.g. patient documentation and pre-discharge education for parents.

Response: Thank you for this comment. We have added the following text in the discussion addressing this point: “On a smaller scale, quality improvement initiatives that seek to improve screening and data capture, educate families about the signs of jaundice, and align clinical care with international guidelines are warranted.”

R2, C8: The references are appropriate and relevant to the topic discussed.

Response: Thank you.

R2, C9: Style suggestions:

L66: -micromol / l or μmol/l as units

L389:-suggest: ..include aspects (or factors) such as …

Response: Thank you for these comments, and we have taken the suggestions and revised accordingly.

Reviewer 3 Report

Comments and Suggestions for Authors

This works reports Neonatal jaundice treatment versus recommendations from BiliApp.

Authors claimed that without rapid diagnostic capability, diagnosis and treatment of Neonatal jaundice is difficult. This hypothesis is also the focus of the script. Although this claim is obvious by common sense but seems not to be supported by the current data and results stated in this manuscript. BiliApp has been proposed as the rapid diagnostic recommendation, results show great discrepancies between recommendation and actual treatment received. For example, table 3, BiliApp suggested no treatment for 36 patients, and exchange therapy for 41 patients. However, in actual treatment, 27 patients were treated, and 20 patients did not go for exchange therapy. The above can be considered as 3 out 4 false-negative recommendations, which can lead to serious complications or even deaths. Thus, hard to demonstrate that rapid diagnostic capability can help the diagnosis and treatment of Neonatal jaundice.    

Author should provide the rationale for using BiliApp as the method to pose as the rapid diagnostic recommendations, rather than other strategies and methodologies. A more robust rapid methodologies with better accuracy, even if more expensive and not suitable for low-resource region can be used to proof your hypothesis. Moreover, a smaller scale pilot test can be performed for feasibility testing prior to running a retrospective test for a year.

Please standardize the data formats and result representations, some percentages are not correct. For example, table 1, maternal age 15-19, overall sample 63 (5.9); 5.9 means the percentage of 63 out of 1059. However, for maternal education none, overall sample 48 (5.2); 5.2 here means 48 out of 923 not 1059. Moreover, please indicate that numbers in brackets are percentages relative to 1059 (or N, please specify).

Please give explanations and details for the missing/incomplete data from the 1059 subjects in this study.

The position of heading for conclusion is not correct.

These above points should be revised and addressed, otherwise, this article is hard to be publishable. thank you very much for your understanding.

Author Response

REVIEWER 3

This works reports Neonatal jaundice treatment versus recommendations from BiliApp.

R3, C1: Authors claimed that without rapid diagnostic capability, diagnosis and treatment of Neonatal jaundice is difficult. This hypothesis is also the focus of the script. Although this claim is obvious by common sense but seems not to be supported by the current data and results stated in this manuscript. BiliApp has been proposed as the rapid diagnostic recommendation, results show great discrepancies between recommendation and actual treatment received. For example, table 3, BiliApp suggested no treatment for 36 patients, and exchange therapy for 41 patients. However, in actual treatment, 27 patients were treated, and 20 patients did not go for exchange therapy. The above can be considered as 3 out 4 false-negative recommendations, which can lead to serious complications or even deaths. Thus, hard to demonstrate that rapid diagnostic capability can help the diagnosis and treatment of Neonatal jaundice.

Response: Thank you for your careful review of this study. Our study used the Bili-App in hindsight to determine how well the treatments provided aligned with NCE guidelines, based on serum bilirubin drawn at admission and reported 1-2 days later. This was not an assessment of how well the Bili-App served as a rapid diagnostic tool – instead it was an assessment of how well treatment patterns aligned with the Bili-App’s recommendations in retrospect. Given the clinical setting, with a nurse to patient ratio of 11:1, given slow turn-around time for serum blood tests, given patients refusal to pay for serum blood tests, it is not surprising that presumptive treatment was sometimes initiated for newborns who might not have needed it, or that newborns with more severe illness were not identified within the study window.

R3, C2: Author should provide the rationale for using BiliApp as the method to pose as the rapid diagnostic recommendations, rather than other strategies and methodologies. A more robust rapid methodologies with better accuracy, even if more expensive and not suitable for low-resource region can be used to proof your hypothesis. Moreover, a smaller scale pilot test can be performed for feasibility testing prior to running a retrospective test for a year.

Response: Thank you for your careful review of this manuscript. Perhaps we were not clear in our study goals, and as such we have attempted to clarify the language. This was not a study designed to test the BiliApp as a rapid diagnostic tool, but rather a study to determine how closely aligned treatment offered at a district hospital in Ghana was to the UK NCE Guidelines. 

R3, C3: Please standardize the data formats and result representations, some percentages are not correct. For example, table 1, maternal age 15-19, overall sample 63 (5.9); 5.9 means the percentage of 63 out of 1059. However, for maternal education none, overall sample 48 (5.2); 5.2 here means 48 out of 923 not 1059. Moreover, please indicate that numbers in brackets are percentages relative to 1059 (or N, please specify).

Please give explanations and details for the missing/incomplete data from the 1059 subjects in this study.

Response: Thank you for your careful review of our data tables. We have included the N next to each variable for which it was not 1059, and we have indicated how many data points were missing for each variable.  We have also added the following text to the limitations: “Missing data from paper charts was a challenge, and despite working hard to track down values for every variable, our dataset did suffer from inconsistent Ns related to missing data. With the exception of maternal education and newborn feeding type, both of which had high levels of missing data, missingness hovered around 1% of total values.”

R3, C4: The position of heading for conclusion is not correct.

Response: Thank you for your careful review. We have revised accordingly.

R3, C5: These above points should be revised and addressed, otherwise, this article is hard to be publishable. thank you very much for your understanding.

Response: We have addressed the concerns raised by Reviewer 3, and in light of the favorable reviews of Reviewers 1, 2, and 4, we believe the manuscript is suitable for publication.

Reviewer 4 Report

Comments and Suggestions for Authors

This is a valuable study of a very important topic for neonatal care in sub-Saharan Africa and other lower resource areas. The manuscript is well structured and presented. Specific recommendations are as follows:

In the Introduction section. It would be helpful to provide a little more physiological detail about the mechanisms underlying neonatal jaundice and the risks it poses. For example it would be helpful to describe briefly the process of haemoglobin breakdown, rise in unconjugated bilirubin levels and the physiological delay in achieving normal conjugation rates in the liver. It should also be made clear that it is unconjugated bilirubin that is the toxic compound for CNS injury as it is capable of crossing the blood-brain barrier. Although most of this is basic background information it cannot be assumed that all readers understand this.  

In the Discussion section it would be helpful to describe alternative novel approaches to bilirubin determination in low-resource settings. Eg BMJ Paediatr Open. 2025 May 8;9(1):e002242. 

Also see PLoS One 2020 Mar 2;15(3):e0216970

https://www.nature.com/articles/s41598-025-89528-6

Author Response

REVIEWER 4

R4, C1: This is a valuable study of a very important topic for neonatal care in sub-Saharan Africa and other lower resource areas. The manuscript is well structured and presented.

Response: Thank you for your careful review of this manuscript.

R4, C2: In the Introduction section. It would be helpful to provide a little more physiological detail about the mechanisms underlying neonatal jaundice and the risks it poses. For example it would be helpful to describe briefly the process of haemoglobin breakdown, rise in unconjugated bilirubin levels and the physiological delay in achieving normal conjugation rates in the liver. It should also be made clear that it is unconjugated bilirubin that is the toxic compound for CNS injury as it is capable of crossing the blood-brain barrier. Although most of this is basic background information it cannot be assumed that all readers understand this.  

Response: Thank you for this comment. We have added text as suggested.

R4, C3: In the Discussion section it would be helpful to describe alternative novel approaches to bilirubin determination in low-resource settings. Eg BMJ Paediatr Open. 2025 May 8;9(1):e002242. 

Also see PLoS One 2020 Mar 2;15(3):e0216970

https://www.nature.com/articles/s41598-025-89528-6

Response: Thank you for this suggestion. We have added text in the discussion to this effect.